# Health-Related Quality of Life during 26-Week Intervention with the New Nordic Renal Diet

**DOI:** 10.3390/nu16132038

**Published:** 2024-06-27

**Authors:** Nikita Misella Hansen, Anne-Lise Kamper, Marianne Rix, Bo Feldt-Rasmussen, Peder Berg, Arne Astrup, Louise Salomo

**Affiliations:** 1Department of Nephrology, Copenhagen University Hospital, Rigshospitalet, Blegdamsvej 9, 2100 Copenhagen, Denmark; 2Department of Clinical Medicine, Faculty of Health and Medical Sciences, University of Copenhagen, 2100 Copenhagen, Denmark; 3Department of Biomedicine, Aarhus University, 8000 Aarhus, Denmark; 4Department of Obesity and Nutritional Sciences, Novo Nordisk Foundation, 2900 Hellerup, Denmark

**Keywords:** chronic kidney disease, health-related quality of life, EQ-5D-5L, Likert scale, dietary intervention, randomized clinical trial, phosphorus, food delivery

## Abstract

The New Nordic Renal Diet (NNRD) is a whole-food approach, tailored to meet recommended guidelines in patients with moderate chronic kidney disease (stage 3b–4). The NNRD improved various metabolic and physiological endpoints during a 26-week randomized controlled study. Here, we examined the effect of dietary intervention on health-related quality of life (HRQoL). Sixty participants were recruited (NNRD group *n* = 30, control group *n* = 30) and 58 completed the study. During the intervention, the NNRD group received food boxes, and recipes once a week. The control group continued their habitual diet. HRQoL was examined at baseline and at the end of the intervention using the validated EuroQol-5D-5L, including a 5-point scale Likert questionnaire at the end of the intervention. Assessed by the EuroQol-5D-5L questionnaire, the NNRD group experienced a reduction in pain/discomfort during the intervention by 26% [−0.44 points (95% CI; −0.73, −0.16)], compared with no change in the control group [0.25 points (95% CI; −0.02, 0.53)] and a between-group difference of −0.70 points (95% CI; −1.03, −0.37, *p* < 0.001). A larger decrease of body fat mass was associated with a larger decrease in pain/discomfort (*p* = 0.014). In addition, the NNRD group reported an overall improvement in conducting usual daily activities by 23% [−0.30-point (95% CI; −0.50, −0.11)], while no change was seen in the control group [−0.02 points (95% CI; −0.21, 0.17)], with a between-group difference −0.28 points (95% CI; −0.51, −0.06, *p* = 0.014). A larger decrease in 24 h urine phosphorus excretion, used as a marker of compliance, was associated with a larger improvement in conducting usual daily activities (*p* = 0.036). The NNRD group had a clinically relevant improvement in various HRQoL outcomes.

## 1. Introduction

Chronic kidney disease (CKD) is associated with reduced health-related quality of life (HRQoL), when compared with the general population [1,2,3]. A variety of complications might contribute to a reduction in HRQoL. Fatigue is a well-known complication to CKD that can affect patients’ lives in numerous ways [4]. Complications such as loss of appetite and itchy skin also occur as CKD progresses [5], which could potentially affect HRQoL. HRQoL is an important tool by which to measure patients’ own perspectives on their physical, mental, and social health status, and the impact on their quality of life [6,7]. Thus, it is important to assess HRQoL in patients with CKD, as studies indicate that HRQoL is already reduced in the early stages of the disease, and deteriorates as the disease progresses [8,9]. Furthermore, HRQoL has been shown to be an independent risk factor for mortality in CKD patients [8,10,11] and further studies indicate that reduced HRQoL is negatively correlated to glomerular filtration rate (GFR) [12,13,14].

Our group has created the New Nordic Renal Diet (NNRD), a whole food approach tailored to patients with moderate CKD. The NNRD has been tested in a 26-week randomized trial, on 60 patients with a mean estimated glomerular filtration rate (eGFR) of 34 mL/min/1.73 m^2^. As part of the study, the intervention group was asked to change their dietary habits to fit the NNRD. When compared with a habitual diet, we found that NNRD reduced phosphorus excretion and proteinuria and caused a relevant weight loss, mainly by loss of fat mass, as well as a reduction in systolic blood pressure [15]. Studies have previously concluded that a healthy diet is associated with better health perception [7], and dietary quality has been shown to be one of the strongest determinants of HRQoL [7,16]. According to international guidelines, a healthy renal diet for patients with moderate CKD focuses on a limited amount of protein (0.6–0.8 g/kg/day), a reduced amount of sodium (<2.3 g/d), and a controlled amount of phosphorus and potassium according to laboratory values [17]. Traditionally, renal nutrition has been characterized by being restricted and primarily focused on single nutrients as opposed to a complete diet. The aim of the NNRD was to offer patients with moderate CKD a healthy diet tailored to meet all nutritional needs [15]. 

As part of the original study, HRQoL, which is a part of patient-reported outcome measures (PROMs), were examined. The use of PROMs are known to contribute valuable information from the patient’s own perspective [5,18]. Thus, PROMs are intended to cover aspects such as quality of life, functional status, and personal experiences among others [18,19]. 

When interpretating PROMs, it is relevant to consider the minimal clinical important differences that are used to assess the effects of the intervention in relation to the outcome from the perspectives of both patients and clinicians [20,21].

In previous studies, NNRD has been shown to be feasible and effective [15,22]. However, the relationship between the NNRD and overall HRQoL has yet to be explored. Therefore, the objective of this study was to explore PROMs by assessment of HRQoL during the 26-week study period.

## 2. Materials and Methods

In-depth details about this study have been published [15]. The study was registered at www.ClinicalTrials.gov (registry number: NCT04579315) and approved by the Danish Capital Region Ethics Committee (H-20026376). The study was carried out in accordance with the Helsinki Declaration.

### 2.1. A Summary of the Study Design

In summary, a non-blinded randomized controlled study was carried out at the Department of Nephrology, Copenhagen University Hospital, between November 2020 and May 2022. The purpose of this study was to explore the health effects of the NNRD in 60 patients with CKD stage 3–4 in a 26-week study period. Thirty participants were randomized to habitual diet, with no intervention (control group), and thirty participants were randomized to a dietary intervention (NNRD group). Participants were eligible to take part in the study if they were >18 years of age, had an eGFR of 20–45 mL/min/1.73 m^2^ (stage 3b–4), and had stable medical treatment for two months prior to initiation of the study. Additionally, they had to comprehend written and spoken Danish. Participants were ineligible for the study if they received phosphate binder treatment, if they had any metabolic disorders that necessitated special dietary restrictions, or if they had received chemotherapy within the previous 6 months. Women who were pregnant or breastfeeding were also excluded. Finally, vegans and participants with food allergies were not eligible to participate in the study. Participants were randomized after informed consent was obtained.

During the 26-week intervention period, the NNRD group received food boxes with all necessary food items and recipes each week, free of charge. The food box contained fresh food items for each meal as well as recipes for five days of the week; the remaining two days, they prepared their meals using the NNRD food principles. Participants were invited to include their household in the food delivery (at their own cost), 15 participants included their families [15].

Regardless of randomization, all participants were subjected to the same examinations during the seven control visits in the 26-week intervention period. Examinations took place at the Department of Nephrology, Copenhagen University Hospital. Fasting blood samples, 24 h urine collections, and 48 h dietary record were all collected. All control visits during the intervention period were identical, with the exception of the first control visit, which took place two weeks after the study began. This visit consisted solely of 24 h urine collections and fasting blood samples. As part of these visits all participants had standard tests performed routinely, e.g., eGFR, plasma creatinine, urinary phosphorus excretion and proteinuria among others, full description is published in our original study [15]. Additionally, at baseline and at the end of the intervention period, participants completed a questionnaire concerning HRQoL. Moreover, both groups completed an additional questionnaire on HRQoL as part of a 5-point Likert scale at the final study visit. 

### 2.2. A Summary of the NNRD

The NNRD arises from the New Nordic Diet (NND), a Scandinavian alternative to the Mediterranean diet [23]. The NND has been tested in the Danish population and has shown promising outcomes on lipid metabolism, body composition and blood pressure [24]. In accordance with international recommendations for CKD patients, our group modified the NND into the NNRD by reducing the amount of phosphorus, protein, and sodium [17]. The NNRD was created to meet the needs and preferences of Nordic CKD patients and is characterized by consisting of 80% plant-based food products and 20% products of animal origin [25]. In brief, the NNRD contained 850 mg phosphorus/day, 30–35 kcal/day, 0.8 g protein/kg/day, and a maximum of 5 g sodium chloride/day. The guidelines for the NNRD are listed in Table 1. 

A full description of the NNRD has previously been published [15].

### 2.3. Health-Related Quality of Life

To assess HRQoL, the participants were asked to answer two questionnaires.

At first, both groups completed the validated five-dimension EuroQol-5D-5L (EQ-5D-5L) questionnaire at baseline, and end of the study [26,27]. This questionnaire covers the following topics: mobility, self-care, usual activities, pain/discomfort, anxiety/depression, and a self-rated score regarding their own interpretation of health on the specific day the questionnaire was answered. The participants were asked to rate their responses on a scale from 1 to 5, with 1 point representing the best possible outcome and 5 points representing the worst possible outcome. An exception is the final question in the EQ-5D-5L questionnaire, which gives a score on the basis of a visual analogue scale (VAS). For this specific question, a higher score was associated with a better outcome, while a lower score was associated with a worse outcome.

Additionally, at week 26, participants from both groups filled in a 5-point Likert questionnaire regarding self-reported HRQoL. The questionnaire contained six questions. For each question, participants were asked to compare their current status to their status prior to the start of the study. Topics included the following: overall health impressions, energy level, mood, abdominal issues, quality of sleep, and changes in skin appearance. Patients could rate their answers by the following: “much better, in some cases slightly better, no change, slightly worse, or much worse.” (Likert scale questionnaire is shown in the Appendix A).

As for the EQ-D5-DL questionnaire, a score of 1 point was associated with the best possible outcome, and a score of 5 points was associated with the worst outcome. 

### 2.4. Statistical Analysis

Statistical analysis was performed using R studio version 4.1.0 and Stata17 for Mac (StataCorp, College Station, TX, USA). Baseline characteristics are presented as mean ± SD.

When using the Likert scale at week 26 to assess the differences between the two groups, a two-sample t test was conducted, values are presented as mean and 95% CI, with corresponding *p* values.

Longitudinal changes of the EQ-5D-5L questionnaire were assessed using linear mixed-effects models for repeated measures. Models estimated the changes from baseline to follow-up at week 26 in the NNRD and control group adjusted for baseline measures of the outcome of interest.

Associations between changes in body fat mass, phosphorus excretion, fractional phosphorus excretion and changes in EQ-5D-5L questionnaires were assessed by simple linear regression analyses and Pearson’s correlation coefficient. Outcomes were reflected in EQ-5D-5L questionnaire scores. Models included visit, change in body fat mass, 24 h urine phosphorus excretion, fractional phosphorus excretion, visit by change in body fat mass, 24 h urine phosphorus excretion, and fractional phosphorus excretion interaction as fixed effects and baseline measures of the outcome of interest as covariate.

A two-sided *p* value < 0.05 was considered statistically significant.

## 3. Results

Sixty participants were recruited to the study. Fifty-eight participants completed the 26-week intervention period, as one participant from each group was excluded due to their undertaking of dialysis treatment, and there were no dropouts. Table 2 displays baseline characteristics (detailed baseline data are published in the original study) [15]. Mean age was 54 years and 55% were females (NNRD group), compared with 55 years and 52% females for the control group. There were no differences in eGFR.

### 3.1. A Summary of the Main Findings from Baseline to Week 26

Several metabolic and anthropometric data were obtained during the 26-week study period. This section aims to summarize the main findings. A full overview of all the findings are published in the original study [15].

During the 26 weeks, the NNRD group reduced their 24 h urine phosphorus excretion by 19% [−153 mg (95% CI; −210, −95)], compared with no change in the control group [18 mg (95% CI; −39, 76)], and with a between-group difference of −171 mg (95% CI; −233, −109, *p* < 0.001). The NNRD group also had a reduction in plasma urea corresponding to −1.5 mmol/L, with a between-group difference of −1.4 mmol/L (95% CI; −2.0, −0.7, *p* < 0.001). Plasma albumin increased by 0.8 g/L in the NNRD group, with a between-group difference of 0.7 g/L (95% CI; 0.0, 1.3, *p* = 0.04). There were no changes in eGFR or plasma creatinine [15].

Moreover, for the 24 h urine markers we found a reduction in urine urea among the NNRD group, which corresponded to −78 mmol, with a between-group difference of −65 mmol (95% CI; −90, −41, *p* < 0.001). Proteinuria was also reduced by 39% (−0.33 g/d) in the NNRD group, with a between-group difference of −0.34 g/d (95% CI; −0.52, −0.17, *p* < 0.001). Finally, there was a reduction in urinary sodium excretion by −54 mmol in the NNRD group, with a between-group difference of −66 mmol (95% CI; −84, −48, *p* < 0.001) [15].

In addition to the metabolic markers, several anthropometric data were collected. When comparing baseline and the final visit after 26 weeks, the NNRD presented with a weight loss of −1.7 kg and a between-group difference of −2.0 kg (95% CI; −3.0, −1.0, *p* < 0.001). BMI was reduced by −0.6 kg/m^2^ in the NNRD group, with a between-group difference of −0.6 cm (95% CI; −1.0, −0.3, *p* < 0.001).

Data from DXA scans show a 5.4% loss of fat mass (−1.2 kg) in the NNRD group, with a between-group difference of −0.7 kg (95% CI; −1.6, 0.1, *p* = 0.08). Concurrently, the NNRD group had a reduction of 2.4% lean mass (−1.1 kg) and a between-group difference of −1.0 kg (95% CI; −1.9, −0.2, *p* = 0.02). Finally, waist circumference was reduced by −1.3 cm in the NNRD group, with a between-group difference of −2.1 cm (95% CI; −3.5, −0.7, *p* = 0.003) [15].

### 3.2. Health-Related Quality of Life during the 26-Week Study Using EQ-5D-5L

EQ-5D-5L was used to assess HRQoL at baseline and week 26. The analysis includes a total of 56 participants (NNRD group *n* = 27; control group *n* = 29). Two participants were excluded from the analysis, as they did not complete the questionnaire. Data are shown in Table 3.

At the end of the intervention, the NNRD group reported a reduction in pain/discomfort by 26% [−0.44 points (95% CI; −0.73, −0.16)], while no change was observed in the control group [0.25 points (95% CI; −0.02, 0.53)], with a between-group difference of −0.70 (95% CI; −1.03, −0.37, *p* < 0.001, Table 3). A larger decrease of body fat mass was associated with a larger decrease in pain/discomfort (*p* = 0.014, Figure 1).

Additionally, the NNRD reported an overall improvement in conducting usual daily activities by 23% assessed by the self-reported score [−0.30-points (95% CI; −0.50, −0.11)], while no change was observed in the control group [−0.02 points (95% CI; −0.21, 0.17)], with a between-group difference of −0.28 points (95% CI; −0.51, −0.06, *p* = 0.014, Table 3). A larger decrease in 24 h urine phosphorus excretion was associated with a larger improvement in conducting usual daily activities (*p* = 0.036, Figure 2).

Finally, when using the assessment of overall health by a VAS score, the NNRD group reported an overall improvement of 10% [7.85 points (95% CI; 2.21, 13.49)], vs. no change in the control group [0.04 points (95% CI; −5.53, 5.46)], with a between-group difference of 7.89 (95% CI; 1.36, 14.41, *p* = 0.018, Table 3). No correlations were found for the VAS score (Appendix A).

### 3.3. Health-Related Quality of Life at End of Study Using the Likert Scale

Additionally, at week 26, HRQoL was measured using a 5-point Likert scale. Data are presented in Table 4. When asked to compare their overall health, the NNRD group reported a mean of 1.9 ± 0.8 compared with 2.8 ± 0.4 in the control group, with a between-group difference of −0.97 points (95% CI; −1.30, −0.64, *p* < 0.001). When asked about energy level, answers responded to 2.1 ± 1.0 in the NNRD group, compared with 2.9 ± 0.6 in the control group, with a between-group difference of −0.85 points (95% CI; −1.29, −0.42, *p* < 0.001). In response to overall mood, the NNRD group reported a mean of 2.2 ± 0.9 vs. 2.8 ± 0.4 in the control group, with a between-group difference of −0.60 points (95% CI; −0.99, −0.21, *p* < 0.001). When asked about abdominal issues, the NNRD group reported a mean of 2.2 ± 1.0 vs. 3.1 ± 0.4 in the control group, with a between-group difference of −0.84 points (95% CI; −1.26, −0.42, *p* < 0.001). In response to quality of sleep, the NNRD reported a mean of 2.4 ± 0.8 compared with 2.9 ± 0.4 in the control group, with a between-group difference of −0.48 points (95% CI; −0.85, −0.12, *p* < 0.001). To the final question, regarding subjective interpretation of skin quality, the NNRD group reported a mean of 2.6 ± 0.7 vs. 3.0 ± 0.2 in the control group, with a between-group difference of −0.37 points (95% CI; −0.67, −0.07, *p* = 0.017) (Table 4) (questionnaire is shown in the Appendix A).

## 4. Discussion

This study demonstrates that following the NNRD for 26 weeks contributes to an overall improvement in HRQoL, when measured both by a validated tool and by a self-constructed Likert scale.

There might be several explanations for the improvements in HRQoL. From a metabolic point of view, there are several factors that might be worth taking into consideration. First, during the 26-week intervention, the NNRD group significantly reduced systolic blood pressure by ~5 mmHg, while antihypertensive medication was reduced [15]. One could suggest that a reduction in medication might be associated with better HRQoL.

In our original study, we found a significant reduction in 24 h urine phosphorus excretion by 19%, a 3.5% reduction in fractional phosphorus excretion, as well as a 39% reduction in proteinuria [15]. In the present study, we found that a larger reduction of 24 h urine phosphorus excretion was associated with improvements in usual activities, thereby implying that successful following of the NNRD is associated with improvements in everyday tasks. It could be possible that following a healthy renal diet (NNRD) can contribute to an overall feeling of having more energy, and thus finding it easier to conduct everyday tasks. Another explanation can simply be that this is a random finding. Further studies are needed to test this hypothesis.

Additionally, our findings suggests that a larger reduction in body fat among the participants is associated with a larger decrease in pain and discomfort. This suggests that a reduction of fat mass is beneficial, given that BMI in the NNRD group and in the control group were 25.8 and 26.2 kg/m^2^, respectively. Though inducing weight loss among the NNRD group was not the goal of our study, the mean weight loss of 1.7 kg during the 26-week intervention, corresponding with <2% of their total bodyweight, is generally expected and accepted when changing a diet. There was a significant reduction in waist circumference in the intervention group, suggesting that the weight loss was primarily located to the abdominal area. In addition, the NNRD group also increased their plasma albumin compared with the control group, indicating sufficient protein intake and thus not an association with the risk of sarcopenia. Other studies support our findings. A study by Bustuil R. et al. also used EQ-5D-5L to measure HRQoL among participants with obesity. Similar to our findings, they were able to show an association between obesity and pain/discomfort [28]. These findings have further been supported in a systematic review by Kolotkin and Andersen, which reports an association between obesity and reduced HRQoL as well as an association between weight loss and improved HRQoL [29].

The NNRD has some of the same characteristics as the Mediterranean diet, such as favoring vegetables, fruits, and fish. Adherence to the Mediterranean diet has previously been associated with improved HRQoL after 3 months [30,31,32]. These positive outcomes have been attributed to a high dietary intake of fibers and antioxidants [30,33]. Polyphenols are a group of antioxidants that have shown to be beneficial in the prevention of various diseases but have also been shown to lead to improvements in overall mental health. A recent review by Gantenbein and Kanaka-Gantenbein highlights the overall effects of the Mediterranean diet as an antioxidant [34]. They suggest that the high presence of polyphenols in fruit, vegetables, nuts, (as in NNRD) and wine may have positive effects on the homeostasis of neurotransmitters in the brain [34]. In addition, polyphenol resveratrol has been shown to increase the levels of both dopamine and serotonin in mice [35].

Moreover, studies have shown that regular intake of fish from the diet, or supplementation with fish oil, is directly associated with higher mental wellbeing, both in a healthy population and in people with depression [31,36,37]. It is already well known that polyunsaturated fatty acids are essential and must therefore be supplied from the diet [38]. Omega-3 fatty acids, present in fish, have been suggested to improve markers on mental health by activating neurotransmitters in the brain and via its anti-inflammatory actions [38,39]. During the 26-week intervention with the NNRD, the participants were instructed to eat fish at least two times per week.

Another dietary aspect is the effects of low protein diets, which have previously been examined in relation to HRQoL in CKD patients. A multicenter study included 153 CKD patients from France and compared them to 128 Italian CKD patients, both groups received 0.6–0.8 g protein/kg/day. On the basis of this study, the authors concluded that low protein diets were safe and not associated with a reduction in HRQoL [40]. The present study confirms these findings, as the NNRD group received 0.8 g protein/kg/day which reduced proteinuri by 39% and p-urea by 1.5 mmol/L, while improving HRQoL outcomes [15].

A recent study by Lee et al. (2023), showed conflicting results. Correspondingly, the authors aimed to investigate the effects of a low protein diet (≤0.8 g/kg/day) on HRQoL outcomes in 571 CKD patients. In contrast, they found that a low protein diet was significantly associated with impaired HRQoL and depressive symptoms [41]. In accordance with our study, Lee et al., used the South Korean version of the validated EQ-5D-5L to assess HRQoL. These conflicting results might be explained simply by the differences in the two diets, as well as differences in study designs. In our study, the NNRD was designed to accommodate cultural traditions and be flavorful, thus providing the participants with a sensation of receiving high quality meals, with an emphasis on new ideas and different food choices rather than constraints.

From a psychological point of view, one important aspect to consider is that the participants received weekly home delivery of fresh food and recipes free of charge. These were created in collaboration with publicly acclaimed Danish chefs, and this might have contributed to the flavorful and joyful food experiences of our diet when compared with habitual diets.

The close follow up and in-depth counseling provided by the investigator throughout the study may have strengthened the collaboration between the participants and the study team, which might have increased the participant’s feeling of self-management.

Self-management is an important aspect to consider in patients with chronic diseases. The concept refers to patients’ own responsibilities in the management of the disease, and has been defined as positive efforts in monitoring health and symptoms [42]. The goal of self-management is to identify strategies that patients can use to manage their disease, while living an active life [43]. The primary idea is to create a sense of control through a shift away from passive education, to a more active role, where patients participate in the optimization of their situation [43]. It has previously been demonstrated that self-management in patients with CKD is associated with improvements in HRQoL aspects [42], as well as a reduction in proteinuria and blood pressure [43]. This is of high relevance for our study, in which we have provided the participants with detailed guidance on how to take control over their current state, by instructing them in the methods by which to choose appropriate foods, thus giving the participants tools that they can actively use in all aspects of life.

The design of the study is a strength. However, there are also relevant limitations to highlight. First, blinding was not possible and so the responses to the questionnaires might be biased. While the examinations were underway, the participants were able to routinely monitor their health improvements, which might contribute to a biased subjective impression of overall improvement in HRQoL. There was no control of beverages intake, tobacco, or physical activity. Additionally, the Likert questionnaire was self-constructed by the investigator and only performed once at the end of the intervention, thus comparison over time was not possible. Finally, the participants in the NNRD group received weekly home delivery of fresh food items for 26 weeks. This unique opportunity might have contributed to an overall subjective improvement in HRQoL, without directly being associated with the NNRD, but rather the concept of free food for 26 weeks.

## 5. Conclusions

In conclusion, this study suggests that 26 weeks of intervention with a diet designed to meet the guideline recommendations in CKD is feasible and is associated with an overall improvement of several aspects of HRQoL. The question of whether the NNRD may become an accepted whole-food concept in real-life settings still needs to be explored.

## Figures and Tables

**Figure 1 nutrients-16-02038-f001:**
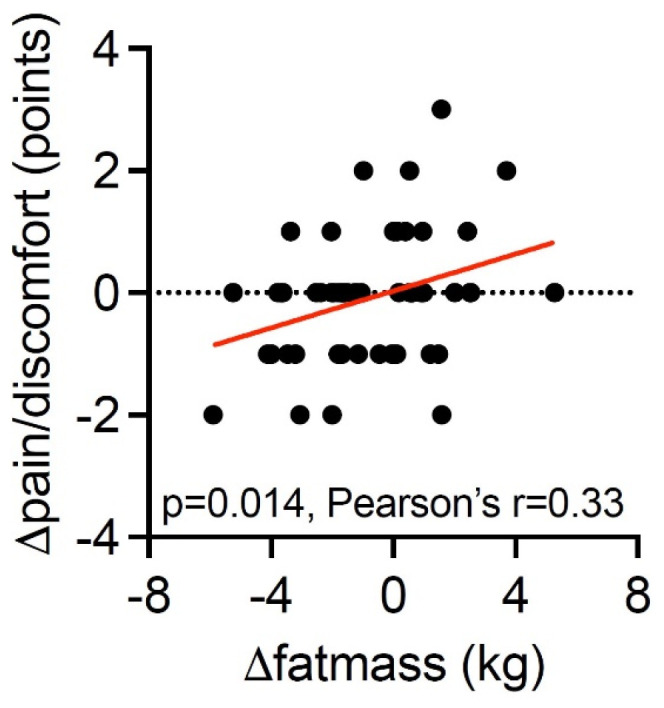
Correlation between change in pain and discomfort according to EQ-5D-5L questionnaire and change in mean body fat measured by DXA scan at baseline and 26 weeks. The red line represents the line of best fit from simple linear regression.

**Figure 2 nutrients-16-02038-f002:**
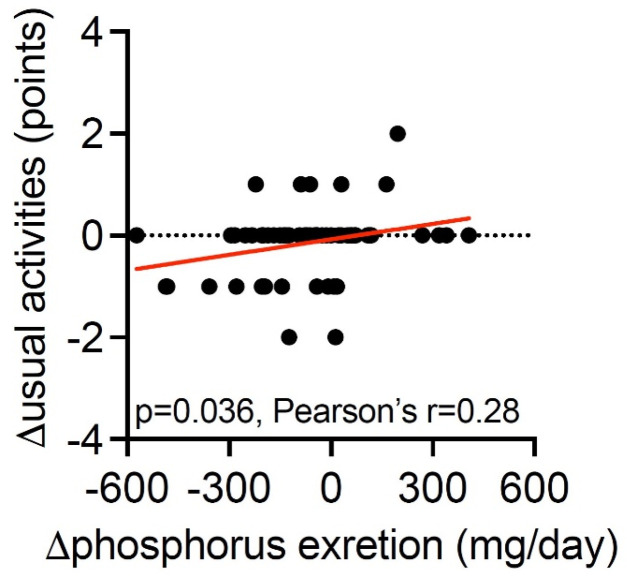
Correlation between change in usual activities according to the EQ-5D-5L questionnaire and change in mean 24 h urine excretion of phosphorus during the total intervention period. The red line represents the line of best fit from simple linear regression.

**Table 1 nutrients-16-02038-t001:** Guidelines of the New Nordic Renal Diet.

Nutrients	Food Items
Energy (Kcal/kg/day): 30–35	80% plant-based, 20% animal origin
Protein (g/kg/day): 0.8	Fruits (g/d): 300
Total fat (E%): <35	Nuts (g/d): 20
Total carbohydrate (E%): 54	Fish intake (servings/week): >2
Added sugar (E%): <10	Poultry (servings/week): >2
Fiber (g/d): 20–30	Red meat (g): 0
Phosphorus (mg/d): 850	No food additives
Sodium chloride (table salt) (g/d): <5	Full vegetarian days (servings/week): minimum 1
Potassium (mg/d): 3000	Egg yolk (servings/week): maximum 2
Calcium (mg/d): 800–1000	At least 95% organic food products
Magnesium (mg/d): 300–370	Flavor enhancement by herbs and spices
	Seasonal orientated
	Easily manageable dishes

**Table 2 nutrients-16-02038-t002:** Baseline characteristics.

	Control ^1^	NNRD ^1^
N	30	30
Age (years)	55 ± 13	54 ± 12
Sex (male/female)	15/15	14/16
Ethnicity (Caucasian/African/Arab/other)	28/1/1/0	27/2/0/1
Cause of CKD (*n*)		
Hypertension	13	10
Polycystic kidney disease	3	4
Glomerulonephritis	5	6
Diabetes mellitus type II	1	2
Other	8	8
Medication (*n*)		
Antihypertensive medication	29	29
Diuretics	11	9
Proton pump inhibitor	6	6
Vitamin D supplement	11	10
Anthropometrics		
Height (cm)	172 ± 10	173 ± 8
Weight (kg)	79.2 ± 16.8	78.4 ± 13
BMI (kg/m^2^)	26.2 ± 4.6	25.8 ± 4.1
Metabolic syndrome	17 (57%)	16 (53%)
Blood markers		
Creatinine (μmol/L)	188 ± 69	173 ± 51
eGFR (mL/min/1.73 m^2^)	33.5 ± 10.2	34.7 ± 8.7
Creatinine clearance (mL/min)	54.8 ± 29.1	55.2 ± 14.8
FGF23 (pg/mL)	192 ± 269	163 ± 117
Phosphate (mmol/L)	1.16 ± 0.22	1.13 ± 0.18
24 h urine markers		
Volume (mL)	2630 ± 1103	2506 ± 738
Phosphorus (mg)	903 ± 354	794 ± 260
Fractional phosphorus excretion (%) ^2^	34.1 ± 9.1	29.4 ± 9.5
Protein (g)	1.09 ± 1.45	0.85 ± 0.99
Urea (mmol)	400 ± 189	360 ± 106
Sodium (mmol)	156 ± 75	166 ± 55

^1^ Mean ± SD for both groups, unless otherwise indicated. ^2^ Fractional phosphorus excretion was calculated as follows: ((Urine phosphorus concentration mmol/L × p-creatinine concentration mmol/L)/(p-phosphorus concentration mmol/L × Urine creatinine concentration mmol/L)) × 100.

**Table 3 nutrients-16-02038-t003:** Changes in health-related quality of life using EQ-5D-5L between the NNRD group and control group (habitual diet) from baseline to 26 weeks.

EQ-5D-5L	Baseline		Change from Baseline (Control vs. NNRD)			
	Control ^1^	NNRD ^1^	Control ^2^	NNRD ^2^	Between-Group Difference ^2^	*p*
Mobility	1.4 ± 0.8	1.3 ± 0.6	−0.03 (−0.21, 0.15)	−0.23 (−0.41, −0.04)	−0.20 (−0.42, 0.01)	0.06
Selfcare	1.1 ± 0.6	1.0 ± 0.0	−0.07 (−0.16, 0.02)	−0.03 (−0.12, 0.06)	0.04 (−0.07, 0.15)	0.46
Usual activities	1.5 ± 0.9	1.3 ± 0.6	−0.02 (−0.21, 0.17)	−0.30 (−0.50, −0.11)	−0.28 (−0.51, −0.06)	0.014
Pain/discomfort	2.0 ± 0.9	1.7 ± 0.9	0.25 (−0.02, 0.53)	−0.44 (−0.73, −0.16)	−0.70 (−1.03, −0.37)	<0.001
Anxiety/depression	1.8 ± 1.0	1.4 ± 0.3	−0.07 (−0.32, 0.17)	−0.22 (−0.47, 0.03)	−0.15 (−0.44, 0.14)	0.31
Overall health score (EQ_VAS)	71.0 ± 21.3	77.9 ± 12.9	−0.04 (−5.53, 5.46)	7.85 (2.21, 13.49)	7.89 (1.36, 14.41)	0.018

^1^ Values are presented as mean ± SD. ^2^ Values are presented as mean (95% CI).

**Table 4 nutrients-16-02038-t004:** Likert scale on health-related quality of life between the NNRD group and control group (habitual diet) at the final study visit (26 weeks).

	Control ^1^	NNRD ^1^	Between-Group Difference ^2^	*p*
HRQoL1 (Overall)	2.8 ± 0.4	1.9 ± 0.8	−0.97 (−1.30, −0.64)	<0.001
HRQoL2 (Energy)	2.9 ± 0.6	2.1 ± 1.0	−0.85 (−1.29, −0.42)	<0.001
HRQoL3 (Mood)	2.8 ± 0.4	2.2 ± 0.9	−0.60 (−0.99, −0.21)	<0.001
HRQoL4 (Abdominal)	3.1 ± 0.4	2.2 ± 1.0	−0.84 (−1.26, −0.42)	<0.001
HRQoL5 (Sleep)	2.9 ± 0.4	2.4 ± 0.8	−0.48 (−0.85, −0.12)	<0.001
HRQoL6 (Skin)	3.0 ± 0.2	2.6 ± 0.7	−0.37 (−0.67, −0.07)	0.017

^1^ Values are presented as mean ± SD ^2^ Values are presented as mean (95% CI).

## Data Availability

The data supporting this study’s findings are available on request from the corresponding author (N.M.H.). The data are not publicly available due to ethical considerations.

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
