# Peer review of "Health-Related Quality of Life during 26-Week Intervention with the New Nordic Renal Diet"

_nutrients, 2024, doi:10.3390/nu16132038_

Round 1

Reviewer 1 Report

Comments and Suggestions for Authors

This study focused on the outcomes of the Nordic Renal Diet compared to a traditional diet over the course of 26-weeks. As aspects of this study were published recently, it appears that the outcomes to be focused on was about the results of the questionnaire. Some aspects to strengthen this manuscript include:

Abstract: Please indicate which stage of kidney disease or eGFR when mentioning moderate in the first sentence. The second sentence was part of a previous study or this study? If so, may consider placing this more in the conclusion part of the abstract. Please include some of the statistical tests used. Beyond just mentioning about the questionnaire, do include the other physiological/metabolic analyses in the methods to coincide with what is reported in the results. For a conclusion, may consider the practical implications of this study.

Introduction:

The second paragraph and beyond appeared to focus more on the study design and the methods. However, there should be more about the current guidelines, why this diet needed to be designed for those with CKD, and more about the population at large regarding their normal consumption patterns, prevalence of CKD in this country, etc. There also needs a purpose statement and the hypothesis and not the overview of the study.

Methods:

As only the statistical analysis portion discussed analysis with body mass and phosphorus excretion, please include this information earlier in the methods about the frequency of collection and rationale for phosphorus excretion and not other values (e.g. potassium/sodium, protein).

Results:

In the first table there is results regarding protein, urea, etc, this needs to be included in the methods, even if there was a lengthier description that has already been published.

Discussion:

There was mention about a reduction in blood pressure, but this was not included in the results nor the methods. If blood pressure was measured, include this into both sections or else remove from the discussion. There was mention about the phosphorus, but there could have been elaboration on how this mechanism would impact perception of quality of life.

Author Response

This study focused on the outcomes of the Nordic Renal Diet compared to a traditional diet over the course of 26-weeks. As aspects of this study were published recently, it appears that the outcomes to be focused on was about the results of the questionnaire. Some aspects to strengthen this manuscript include:

Abstract: Please indicate which stage of kidney disease or eGFR when mentioning moderate in the first sentence.
This information has been included and specified to stage 3b-4.

The second sentence was part of a previous study or this study? If so, may consider placing this more in the conclusion part of the abstract.
Please include some of the statistical tests used. Beyond just mentioning about the questionnaire, do include the other physiological/metabolic analyses in the methods to coincide with what is reported in the results. For a conclusion, may consider the practical implications of this study.
This information is related to our original study that has previously been published. The goal of this paper is solely to investigate the health-related quality of life. We have included more details from the original study in the methods and result sections.

Introduction:

The second paragraph and beyond appeared to focus more on the study design and the methods. However, there should be more about the current guidelines, why this diet needed to be designed for those with CKD, and more about the population at large regarding their normal consumption patterns, prevalence of CKD in this country, etc. There also needs a purpose statement and the hypothesis and not the overview of the study.
We appreciate your suggestion. However, this paper aims to solely focus on the challenges related to health-related quality of life among CKD patients. The goal is not to describe the New Nordic Renal Diet and hence the guidelines for CKD patients, as we have already published all these details in our main study that we are referring to throughout the paper. Nevertheless, we do acknowledge your opinion and have tried our best to include additional information about the dietary guidelines etc. in our introduction, for further information we refer to our original study and the international guidelines.

Methods:

As only the statistical analysis portion discussed analysis with body mass and phosphorus excretion, please include this information earlier in the methods about the frequency of collection and rationale for phosphorus excretion and not other values (e.g. potassium/sodium, protein).
As we have now included several data from our primary study (blood- and urine analyses etc.) we believe that it is relevant to keep the information in its present form.

Results:

In the first table there is results regarding protein, urea, etc, this needs to be included in the methods, even if there was a lengthier description that has already been published.
More details about blood- and urine samples have been included in the method section.

Discussion:

There was mention about a reduction in blood pressure, but this was not included in the results nor the methods. If blood pressure was measured, include this into both sections or else remove from the discussion. There was mention about the phosphorus, but there could have been elaboration on how this mechanism would impact perception of quality of life.
This information is part of our main study. It is mentioned in the discussion as part of an overall discussion of the NNRD. This is not part of the present paper, that solely focus on HRQoL, but in order to make conclusions about the NNRD we find it relevant to include results from other papers.

Reviewer 2 Report

Comments and Suggestions for Authors

This study was aimed to examine the effect of the dietary intervention (the New Nordic Renal Diet; NNRD) on health-related quality of life (HRQoL). As the results, this study demonstrates that following the NNRD for 26-weeks contributes to an overall improvement in HRQoL. The reviewer considers that the results of this study provide very useful information that can contribute to improving the quality of life of patients with chronic kidney disease through dietary intervention (the NNRD). However, there are several questions in this study.

1.     Did the dietary intervention (the NNRD) in this study improve renal function (eGFR or proteinuria) in CKD patients? Moreover, was the improvement in HRQoL associated with improvement in renal function?

2.     Figure 1 shows that a larger decrease of body fat mass was associated with a larger decrease in pain/discomfort. The reviewer believes that in CKD patients, the goal should be to maintain and increase skeletal muscle mass rather than to reduce body fat mass in order to prevent the progression of frail and sarcopenia. Did the authors measure skeletal muscle mass in this study? If so, how did it change before and after the intervention? Furthermore, was the change in skeletal muscle mass before and after the intervention associated with changes in HRQoL?

3.     Figure 2 shows that a large decrease in 24h urine phosphorus excretion was associated with a larger improvement in conducting usual daily activities. However, there is no discussion of this result in the “Discussion” section. Why does increasing 24h urinary phosphorus excretion improve HRQoL?

4.     In the “Discussion” section, the authors discuss the effect of changing each nutrient intake on HRQoL. The results of this study do not show changes in energy intake or each nutrient intake before and after the intervention.

5.     How did the authors manage the non-dietary food intake of the dietary intervention group in this study? In this study, how did the authors manage intake of recreational foods (e.g., coffee, juice, alcohol), supplements, and tobacco?

6.     This study consisted of two groups; a dietary intervention group and control group. As shown in Table 1, the two groups were uniform in terms of age, gender, and physical characteristics of the participants. The authors likely stratified participants before randomly assigning them to two groups, but this manuscript does not describe this stratification.

7.     In this study, sixty CKD patients participated, and two of whom were excluded due to initiation of dialysis. Furthermore, of the 58 participants, 56 were ultimately included in the analysis. What was the reason for the two participants being excluded?

8.     In relation to the above, were there any dropouts among the study participants? Additionally, what was the adherence rate of the dietary intervention group?

Author Response

This study was aimed to examine the effect of the dietary intervention (the New Nordic Renal Diet; NNRD) on health-related quality of life (HRQoL). As the results, this study demonstrates that following the NNRD for 26-weeks contributes to an overall improvement in HRQoL. The reviewer considers that the results of this study provide very useful information that can contribute to improving the quality of life of patients with chronic kidney disease through dietary intervention (the NNRD). However, there are several questions in this study.

  1. Did the dietary intervention (the NNRD) in this study improve renal function (eGFR or proteinuria) in CKD patients? Moreover, was the improvement in HRQoL associated with improvement in renal function?
    The NNRD did not improve renal function (we also did not expect such results in a relatively short interventional period of 26 weeks). We did find a reduction in proteinuria during the intervention. The requested data has already been published in the original study but have now also been included in the result section.
    As for the question about renal function and HRQoL we did not test for association as there were no changes in renal function doing the study.
  2. Figure 1 shows that a larger decrease of body fat mass was associated with a larger decrease in pain/discomfort. The reviewer believes that in CKD patients, the goal should be to maintain and increase skeletal muscle mass rather than to reduce body fat mass in order to prevent the progression of frail and sarcopenia. Did the authors measure skeletal muscle mass in this study? If so, how did it change before and after the intervention? Furthermore, was the change in skeletal muscle mass before and after the intervention associated with changes in HRQoL?
    The goal was never to induce a weight loss among the NNRD group. However, there was a mean weight loss of 1.7 kg during the 26-week intervention (< 2% of their total bodyweight which is generally expected and accepted when changing a diet). There was a significant reduction in waist circumference in the intervention group suggesting the weight loss was primarily located to the abdominal area. In addition, the NNRD group also increased their plasma albumin compared to the control group indicating sufficient protein intake.
    As the mean BMI was 25.8 km/m2, we believe that a mean weight loss of 1.7 kg during a 26-week period is not associated with a risk of sarcopenia. Yet, we do acknowledge that it could have been interesting to perform such analysis, which unfortunately we did not in this present study, as a result we do not have data on skeletal muscle. Data on body composition, e.g. fat mass and lean mass has recently been published in our original study, and have now been included in the result section.
  1. Figure 2 shows that a large decrease in 24h urine phosphorus excretion was associated with a larger improvement in conducting usual daily activities. However, there is no discussion of this result in the “Discussion” section. Why does increasing 24h urinary phosphorus excretion improve HRQoL?
    The following statement is already included in our discussion: “In the present study, we found that a larger reduction of 24h urine phosphorus excretion was associated with improvements in usual activities, thereby implying that following the NNRD is associated with improvements in everyday tasks.” – In other words, based on these results it might be possible that following a healthy renal diet (NNRD) can contribute to an overall feeling of having more energy, and thus finding it easier to conduct everyday tasks. Another explanation can simply be that this is just a random finding. Further studies are needed to test this hypothesis.
  2. In the “Discussion” section, the authors discuss the effect of changing each nutrient intake on HRQoL. The results of this study do not show changes in energy intake or each nutrient intake before and after the intervention.
    There are no statement of this in our discussion. Our discussion focusses on the potential effects of a healthy dietary pattern that are characterized by following the guidelines for patients with moderate CKD. In our primary study we have published these requested data on differences in nutrient intake and energy intake between the two groups, but they are not relevant for this paper, as this paper aims to solely focus on HRQoL. We find it necessary to choose wisely what type of data we republish in this paper, so to avoid republishing a paper that is similar to the original study.
  3. How did the authors manage the non-dietary food intake of the dietary intervention group in this study? In this study, how did the authors manage intake of recreational foods (e.g., coffee, juice, alcohol), supplements, and tobacco?
    Unfortunately, there were no control for these markers.
  4. This study consisted of two groups; a dietary intervention group and control group. As shown in Table 1, the two groups were uniform in terms of age, gender, and physical characteristics of the participants. The authors likely stratified participants before randomly assigning them to two groups, but this manuscript does not describe this stratification.
    The participants who fulfilled the inclusion criteria were randomly assigned to the two groups as described in the methods. There was no stratification to subgroups of any form prior to randomization.
  5. In this study, sixty CKD patients participated, and two of whom were excluded due to initiation of dialysis. Furthermore, of the 58 participants, 56 were ultimately included in the analysis. What was the reason for the two participants being excluded?
    The two participants were excluded as we did not have data as a result of the questionnaires not being completed by the subjects.
  6. In relation to the above, were there any dropouts among the study participants? Additionally, what was the adherence rate of the dietary intervention group?
    The requested information is published in our primary paper that we are referring to. To summarize:
    There were no dropouts. Adherence corresponded to the following: 53% had an average adherence of 75-100%, 23% had an adherence score corresponding to 50%, and 10% had a score that corresponded to not adhering to the diet.

Reviewer 3 Report

Comments and Suggestions for Authors

Dear Authors,

thank you for your work on the manuscript, titled: "Health-related quality of life during 26-week intervention with the New Nordic Renal Diet".

The presented manuscript is a very interesting presentation of the interventional study that the authors performed on CKD patients, stages 3-4. Dietary interventions are of crucial importance in these patients and patient-related outcomes (PROMs) are often underlooked in studies.

Some minor issues have been raised while reading the manuscript:

- due to the eGFR cut-off (20-45 ml/min/1,73 m2) I suggest specifying the inclusion criteria from CKD 3-4 into CKD 3B-4.

- what were the costs in the NNRD group? Do you think the costs of this diet could be an issue?

- what do the authors think about NNRD in other areas and population? Could this diet be applied anywhere else?

Comments on the Quality of English Language

English language is of sound quality.

Author Response

Dear Authors,

thank you for your work on the manuscript, titled: "Health-related quality of life during 26-week intervention with the New Nordic Renal Diet".

The presented manuscript is a very interesting presentation of the interventional study that the authors performed on CKD patients, stages 3-4. Dietary interventions are of crucial importance in these patients and patient-related outcomes (PROMs) are often underlooked in studies.
Some minor issues have been raised while reading the manuscript:

- due to the eGFR cut-off (20-45 ml/min/1,73 m2) I suggest specifying the inclusion criteria from CKD 3-4 into CKD 3B-4.
This has been specified in the abstract and the method section.

- what were the costs in the NNRD group? Do you think the costs of this diet could be an issue?
We have no specific data on the cost of the diet. It can vary a lot, depending to which degree someone would choose to follow the diet. However, the NNRD do encourage organic food products, which can be an additional cost for some households. On the other hand, the NNRD does not recommend intake of red meat etc., which could also be cost effective, in that aspect such dietary changes might outline each other in terms of financial cost, and thereby not be a financial burden for those choosing to follow the NNRD.

- what do the authors think about NNRD in other areas and population? Could this diet be applied anywhere else?
This is a very relevant question. We believe that the NNRD has potential among other population groups. E.g., to serve as a healthy dietary choice for the general population, but the diet could also have great potential among patients with cardiovascular disease and diabetes. Yet, future studies are needed to test these hypotheses.

Round 2

Reviewer 2 Report

Comments and Suggestions for Authors

The reviewer thinks that the authors have responded appropriately to the reviewers' comments. However, the authors' responses were reflected only in their responses to the reviewers and not in the manuscript. Authors are requested to incorporate the content of their responses to the reviewers into their manuscripts.

Author Response

The reviewer thinks that the authors have responded appropriately to the reviewers' comments. However, the authors' responses were reflected only in their responses to the reviewers and not in the manuscript. Authors are requested to incorporate the content of their responses to the reviewers into their manuscripts.

We thank you for your comment. We have now incoorperated all of our responses in the manuscript that was possible for us. Please find the updates in the following sections: Methods, Results and Discussion.